# The Evaluation of a Porcine Circovirus Type 2 (PCV2) Intradermal Vaccine Against a PCV2 Field Strain

**DOI:** 10.3390/vaccines13040343

**Published:** 2025-03-24

**Authors:** Cheng-Kai Hsieh, Chia-Yi Chien, Chun-Wei Liu, Shu-Wei Chang, Hongyao Lin, Leonardo Ellerma, Ming-Tang Chiou, Chao-Nan Lin

**Affiliations:** 1Department of Veterinary Medicine, College of Veterinary Medicine, National Pingtung University of Science and Technology, Pingtung 91201, Taiwan; ps891111@gmail.com (C.-K.H.); a8033376159483@gmail.com (C.-Y.C.); f841019@gmail.com (C.-W.L.); 2Intervet Animal Health Taiwan Ltd., Taipei 11047, Taiwan; hikouki767777@gmail.com; 3MSD Animal Health Innovation Pte Ltd., Singapore 718847, Singapore; hongyao.lin@msd.com; 4MSD Animal Health (Phils.), Inc., Makati City 1226, Philippines; leonardo.ellerma@merck.com; 5Animal Disease Diagnostic Center, College of Veterinary Medicine, National Pingtung University of Science and Technology, Pingtung 91201, Taiwan; 6Research and Technical Center for Sustainable and Intelligent Swine Production, National Pingtung University of Science and Technology, Pingtung 91201, Taiwan

**Keywords:** PCV2, subunit vaccines, heterologous challenge, cross-protection

## Abstract

**Background/Objectives:** Porcine circovirus type 2 (PCV2) has a major impact on swine productivity. Vaccines are used to aid in control and mitigate production losses. We investigated the protection provided by an intradermal PCV2 vaccine against a field strain in Taiwan. **Methods:** We conducted a safety and efficacy study. In the safety study, four Specific Pathogen Free (SPF) piglets were enrolled in the study. One was selected as the control and left unvaccinated, one was selected to be intradermally vaccinated with five times the standard dose (1 mL, Porcilis^®^ PCV ID), and the other two were vaccinated with two times the standard dose (0.4 mL, Porcilis^®^ PCV ID). All animals were observed for 3 weeks for adverse events post-vaccination. In the efficacy study, twelve SPF pigs negative for the PCV2 antibody were randomly divided into two groups. The first group of six pigs was vaccinated (Porcilis PCV ID, 0.2 mL) intradermally at 3 weeks of age. The second group of six pigs was sham vaccinated with 0.2 mL of normal saline. At 7 weeks of age, all pigs were challenged with the PCV2 strain CYC08 (1 × 10^5^ TCID_50_/mL) by nasal and intramuscular injection. Clinical monitoring of body temperature and mortality was conducted daily. At 11 weeks of age, all animals were sacrificed for histopathological analysis. **Results:** No adverse events were reported in the safety study. In the efficacy study, the vaccinated animals had statistically improved results in the following areas post-challenge: body temperature rise, viremia, virus shedding, mortality, tissue histopathological and microscopic scores. **Conclusions:** The study results support that a one-dose PCV2 vaccine administered intradermally with a needle-free injector is safe and provides protection when challenged with a field PCV2 strain.

## 1. Introduction

Porcine circovirus type 2 (PCV2) is a major etiological agent of the Porcine Respiratory Disease Complex (PRDC) [1]. PCV2 is a non-enveloped, circular, single-stranded DNA virus belonging to the genus *Circovirus*, family *Circoviridae* together with the genus *Cyclovirus* [2]. The term porcine circovirus-associated diseases (PCVADs) is used to denote the entire spectrum of diseases that are associated with PCV2. The common presentation seen in the field is subclinical disease, characterized by decreased average daily gain (ADG) and increased feed conversion ratio (FCR). In severe cases or if co-infections with other respiratory pathogens occur, the postweaning multisystemic wasting syndrome (PMWS) can be observed. This is clinically characterized by wasting, respiratory disease and enteritis [1]. These diseases are associated with enormous economic losses in the pig-producing industry [1]. For instance, Alarcon et al. [3] estimated the costs of PCVAD in the UK as ranging from GBP 84.1 to GBP 8.1 per pig, depending on the severity of the infection. In the United States, the disease has cost the pig-producing industry an average of USD 3–4 per pig with peak losses in severe infection up to USD 20 per pig [4].

To mitigate the economic losses of PCVAD, vaccines have been developed and are widely used by producers. PCV2 is currently classified into eight different genotypes, designated from PCV2a to 2h [5]. Over time, the predominant genotypes of PCV2 have slowly shifted from PCV2a to PCV2b to PCV2d [6]. Despite this shift, PCV2d does not escape the immunity induced by PCV2a-based vaccines [7,8]. Commercial PCV2 vaccines have been shown to induce both cellular and humoral immunity [9,10,11,12]. In field trials of PCV2 commercial vaccines, an increase in the average daily gain (ADG) and decrease in mortality rate in vaccinated animals have been demonstrated [13]. Continued PCV2 vaccination can substantially reduce the viral shedding from infected animals, and hence greatly reduce the viral prevalence on the pig farm [14,15].

Swine vaccines are commonly administered through intramuscular (IM) injection [16] via sterile stainless steel needles. Several issues can arise with IM injection. A study conducted in the UK in 2022 [17] found that 81% of farmers were reusing needles with differing frequency of needle changes between animals, raising the possibility of iatrogenic transfer of disease between animals. Porcine reproductive and respiratory syndrome virus (PPRSV) has been shown to spread to susceptible pigs via contaminated needles [18,19]. The same has been demonstrated for African swine fever virus [20] and PCV2 [21]. Secondly, the study of Owen et al. [17] found substantial damage to the needle after 12 uses, leading to increased skin shearing that requires higher force to puncture the skin, and consequentially additional local trauma and inflammation post-vaccination. This potentially contributes to both swine welfare as well as meat quality issues, particularly if needles are reused [22,23]. For instance, the incidence of head and neck abscesses in pigs at slaughter due to injection site lesions that are directly associated with needle reuse has been estimated at 2.51% over a 3-month period [24], representing a substantial economic loss to producers of the most valuable parts of the carcass. A separate study [23] also found that pig carcass defects resulting from the use of hypodermic needle injection range from 2.7% hip bruises to 11.2% neck lesions. Finally, food safety issues exist when broken needles are left in swine carcasses. Broken needles can lead to metal fragments in meat and meat products. To prevent this, many farm procedures commonly call for the needless euthanasia of healthy animals [25].

An alternative to IM vaccination is intradermal (ID) vaccination. Needle-free devices (NFDs) have been used in human medicine to deliver the antigen into the skin through high pressure jets [26]. Presently, vaccines delivered via NFDs are commercially available in pigs, and intradermal vaccination of pigs has been shown to be able to trigger cellular and humoral immune responses, even in the absence of adjuvant [27,28,29,30]. Several commercially available NFDs are specifically designed and tested with ID vaccines, such as IntraDermal Application of Liquids (IDALs) (MSD Animal Health, Rahway, NJ, USA) and Hipradermic (HIPRA, Amer, Girona, Spain). An intradermal vaccine for PCV2 (Porcilis^®^ PCV ID, MSD Animal Health, Rahway, NJ, USA), delivered via the IDAL NFD was made available in the European market in 2016. This vaccine is also recently available in several Asian markets, but there is limited information available around the efficacy of this vaccine against a field strain of PCV2. We therefore designed this study to understand the safety and efficacy of this vaccine in Taiwan, using a local strain as the challenge material.

## 2. Materials and Methods

The animal experimental procedure was reviewed and approved by the Institutional Animal Care and Use Committee (IACUC) of the National Pingtung University of Science and Technology (NPUST), with approval number NPUST-110-115. The vaccine tested in both the safety and efficacy trials was Porcilis^®^ PCV ID (MSD Animal Health, Rahway, NJ, USA) and 10 vials were used (0.2 mL per dose, batch ID number A058A02).

### 2.1. Safety Study

Four 3-week-old Specific Pathogen Free (SPF) piglets were enrolled in the study. One was selected as the control and left unvaccinated, one was selected to be intradermally vaccinated with 5× the standard dose (1 mL, Porcilis^®^ PCV ID) and the other two were vaccinated with 2× the standard dose (0.4 mL, Porcilis^®^ PCV ID). All animals were observed for 3 weeks for adverse events post-vaccination. Specifically, body temperature was measured at 0/4/8/24 h/48 h post-vaccination and the injection site was monitored for local reactions. Pigs were also monitored for general well-being or ill-thrift. Vaccination was conducted intradermally with IDAL^®^ 3G vaccination device (MSD Animal Health, Rahway, NJ, USA).

### 2.2. Efficacy Study

In the efficacy study, a total of 12 SPF pigs negative for the PCV2 antibody were randomly divided into two groups. The first group of 6 pigs was vaccinated (Porcilis PCV ID, 0.2 mL) intradermally at 3 weeks of age. The second group of 6 pigs was sham vaccinated with 0.2 mL of saline as a control group. At 7 weeks of age, all pigs were challenged with the PCV2 strain CYC08 (1 × 10^5^ TCID_50_/mL) by nasal (1 mL) and intramuscular (1 mL) injection. This is a local PCV2a strain in Taiwan that has been demonstrated to be of high virulence [31]. Clinical monitoring of body temperature and mortality was conducted daily. At 11 weeks of age, all animals were sacrificed for histopathological analysis.

#### 2.2.1. Porcine Circovirus Type 2 (PCV2) Antibody Titres

Blood samples were collected from the immunized group and the control group of pigs once a week before immunization, before and after the challenge. Additionally, the PCV2 antibody was detected by a commercial ELISA kit (BioChek B.V., Reeuwijk, Holland, The Netherlands) according to the manufacturer’s instructions, and the antibody S/P value between the immunization group and the control group was compared.

#### 2.2.2. Monitoring of Clinical Parameters and Average Daily Gain

After the challenge, the clinical manifestations (spirit, appetite, excretion, respiration, gait, body surface appearance, etc.) of the immunized group and the control group were observed and recorded twice a day. Additionally, the body temperature was measured, and the mortality, average daily weight gain, and stunted pig incidence of the immunized group and the control group were compared. Stunted pigs refer to pigs that are less than 75% of the average body weight of all pigs at the end of the trial, and the ratio of stunted pigs in the immunization group to the control group is calculated.

#### 2.2.3. Porcine Circovirus Type 2 (PCV2) Virus Shedding Post-Challenge

Nasal and anal samples were taken with a sterile cotton swab, then placed in a sterile phosphate buffered solution, swirled and mixed for at least five seconds, then the swab was discarded, the cap was screwed on, and the relevant information of the sample was recorded, and the sample suspension was stored below −20 °C. Nasal and anal samples were collected before the challenge (0 week post-challenge, 0 WPC), and then 1, 2, 3, and 4 WPC. The viral DNA was extracted from clinical specimens (nasal and anal suspension or serum) as described by Tsai [32]. The samples were tested in accordance to the real-time quantitative PCR (qPCR) method as described by Tsai et al. [32] to detect the presence or absence of PCV2 DNA in stool and nasal samples, and the detection rate and excretion rate of the virus were compared.

#### 2.2.4. Porcine Circovirus Type 2 (PCV2) Serum Viremia Post-Challenge

Blood samples were collected from the immunized group and the control group of pigs once a week before immunization, before and after the challenge. Additionally, the viral load in serum was detected by qPCR [32] and the amount of PCV2 viral nucleic acid in serum at each time point was compared.

#### 2.2.5. Necropsy

All pigs were observed for clinical signs twice a day after vaccination, and their body temperature was measured and recorded. All pigs that did not die within 4 weeks after inoculation to the 4th week were euthanized, and then recorded by autopsy. After the autopsy of pigs, organ tissue sections and staining were interpreted, and the macroscopic pathology and histopathological changes were interpreted by two senior pathological veterinarians, respectively. Additionally, the histopathological changes in the lungs, kidneys, hilar lymph nodes, mesenteric lymph nodes, and groin lymph nodes of the vaccinated group and the control group were compared according to the method of Opriessnig et al. [33].

#### 2.2.6. Immunohistochemical Staining

Paraffin sections of hilar lymph nodes, mesenteric lymph nodes, and groin lymph nodes of the immunized group and the control group were collected for PCV2 immunohistochemical staining [32]. The blind test was performed by two senior pathologists for immunohistochemical staining: ten fields of view were randomly selected at 200 times to observe the strength of the PCV2 antigen antibody response signal of each lymphoid follicle in the field of view, and the PCV2 antigen index was determined with reference to the research report published by Opriessnig et al. [33].

#### 2.2.7. Statistical Analysis

The comparison of PCV2 shedding in anal and nasal swabs was conducted by qPCR, Fisher’s exact test was used to compare the positive rate, and Student’s *t*-test was used to determine the viral load of each group. PCV2 viremia was detected by qPCR, and then the area under the curve (AUC) was calculated by the linear trapezoidal rule, and the Kruskal-Wallis test was used to test between the groups. The total results of histopathological change scores were statistically analyzed by the Wilcoxon rank sum test (with 0.5 continuity correction). The scores of immunohistochemical staining were summed and tested between groups by the Kruskal-Wallis test, and if there were significant differences, Dunn’s multiple comparison method was used to verify the association between groups. Additionally, *p* values < 0.05 and 0.01 were considered statistically significant and highly significant, respectively.

## 3. Results

### 3.1. Safety Study

Three weeks post-immunization, all four pigs were alive and no adverse events were observed. No fever was observed at the 0/4/8/24/48 h time points post-vaccination. A small localized swelling measuring 1–2 cm was observed at the vaccination site in all pigs. This lesion was both localized and mobile, and then disappeared within 1 week of vaccination in all four animals.

### 3.2. Efficacy Study

#### 3.2.1. Clinical Manifestations and Mortality Observation Records and Body Temperature Measurement

At 7 weeks of age, twelve primary SPF pigs were challenged with the PCV2 strain CYC08 (1 × 10^5^ TCID_50_/mL) by 1 mL of nasal injection and 1 mL intramuscularly. The average body temperature of the pigs in the control group reached the temperature of fever (greater than 40.5 °C) on the 12th day after the challenge, and the body temperature of the pigs in the immunization group was normal (Figure 1).

#### 3.2.2. Porcine Circovirus Type 2 (PCV2) Antibody Determination

The PCV2 antibody was detected by commercial ELISA provided by the manufacturer, and both the immunization group and the control group were negative for PCV2-specific antibodies before immunization (at 3 weeks of age) (Figure 2). Before the challenge at 7 weeks of age (4 weeks after immunization or 0 week post-challenge (WPC)), the PCV2-specific antibodies in the immunization group were slightly increased, and the antibody values in the control group were still negative. At 1 WPC, the PCV2-specific antibodies in the immunization group were all positive, and the antibody values in the control group were still negative. At 2 WPC, the PCV2-specific antibody price in the immunized group increased, while the pigs in the control group were still negative (began to increase a little, but still below the threshold), and the antibody price in the immunized group was significantly higher than that in the control group from the time of challenge (4 weeks after immunization or 0 WPC) to the end of the test (4 WPC) (Figure 2).

#### 3.2.3. Comparison of Clinical Observations, Mortality, Average Daily Gain, and Incidence of Stunted Pigs

The clinical symptoms of the pigs in the two groups were observed every day after the challenge. One pig in the control group showed depression, poor appetite, and abdominal breathing at 25 days post-challenge, and the pig died on the morning of sacrifice (4 WPC). On necropsy, gross findings were consistent with polyserositis and enlarged lymph nodes. Bacterial culture from lungs, pericardial and pleural fluid yielded no significant findings. Table 1 shows the comparison of the body weight, average daily gain, mortality, and incidence of stunted pigs between the two groups during the trial period as well as the average daily gain from the challenge to the end of the trial (7–11 weeks old). The average daily gain of the vaccine group (0.377 kg) was significantly higher than that of the control group (0.250 kg) (*p* = 0.0043) (Table 1). During the trial, the mortality rate of the control group was 16.67% (1/6), and that of the vaccine group was 0% (0/6). The incidence of stunted pigs was 16.67% in the control group, while there were no stunted pigs in the vaccine group (Table 1).

#### 3.2.4. Nasal and Fecal Shedding of Porcine Circovirus Type 2 (PCV2)

The PCV2 load of nasal swabs is summarized in Figure 3A. At 1 WPC, the nasal swabs of the two groups were negative for PCV2 DNA before the challenge, and the mean viral load and standard deviation (Mean ± SD) of the vaccine group were 1.91 ± 0.44 Log_10_ copies/μL, with a positive rate of 100%, and the control group was 2.60 ± 0.75 Log_10_ copies/μL, with a positive rate of 100%. At 2 WPC, the average viral load in the vaccine group was 3.18 ± 0.95 Log_10_ copies/μL, with a positive rate of 100%, and the control group was 4.36 ± 0.67 Log_10_ copies/μL, with a positive rate of 100%, and the nasal swab viral load in the vaccine group was significantly lower than that in the control group (*p* < 0.05). At 3 WPC, the average viral load was 2.12 ± 1.15 Log_10_ copies/μL in the vaccine group, with a positive rate of 66.67%, and 3.24 ± 1.27 Log_10_ copies/μL in the control group, with a positive rate of 100%. At 4 WPC, the average viral load was 3.16 ± 0.90 Log_10_ copies/μL in the vaccine group, with a positive rate of 100%, and the positive rate was 100% in the control group with 3.04 ± 1.41 Log_10_ copies/μL. There was a significant difference in the viral load of nasal swabs between the two groups at 2 WPC, but there was no significant difference in viral load and positivity rate in the rest of the weeks (Figure 3A).

The results of anal swabs are summarized in Figure 3B. The anal swabs of the two groups were negative for PCV2 DNA before the challenge. At 1WPC, the mean viral load and standard deviation (Mean ± SD) of the vaccine group were 1.91 ± 0.46 Log_10_ copies/μL, with a positive rate of 100%, and the control group was 4.09 ± 0.66 Log_10_ copies/μL, with a positive rate of 100%, and the average viral load of anal swabs in the vaccine group was significantly lower than that in the control group (*p* < 0.001). At 2 WPC, the average viral load in the vaccine group was 2.58 ± 1.67 Log_10_ copies/μL, with a positive rate of 83.33%, and the viral load in the control group was 5.92 ± 0.34 Log_10_ copies/μL, with a positive rate of 100%, and the viral load of anal swabs in the vaccine group was significantly lower than that in the control group (*p* < 0.001). At 3 WPC, the average viral load was 2.22 ± 2.01 Log_10_ copies/μL in the vaccine group, with a positive rate of 50%, and the viral load in the control group was 4.50 ± 1.05 Log_10_ copies/μL, with a positive rate of 100%, and the viral load of anal swabs in the vaccine group was significantly lower than that in the control group (*p* < 0.05). At 4 WPC, the average viral load in the vaccine group was 2.69 ± 1.30 Log_10_ copies/μL, with a positive rate of 100%, and the viral load in the control group was 4.42 ± 1.05 Log_10_ copies/μL, with a positive rate of 100%, and the viral load of anal swabs in the vaccine group was significantly lower than that in the control group (*p* < 0.05). From 1 week after the challenge to the end of the trial (4 weeks after the challenge), the viral load of anal swabs in the vaccine group was significantly lower than that in the control group (Figure 3B).

#### 3.2.5. Porcine Circovirus Type 2 (PCV2) Load in Serum

The mean viral load and standard deviation (Mean ± SD) of the vaccine group were 0.95 ± 0.83 Log_10_ copies/μL at 1 WPC, with a positive rate of 66.67%, and that of the control group was 5.14 ± 0.98 Log_10_ copies/μL, with a positive rate of 100% (Figure 4). At 2 WPC, the average viral load was 0.46 ± 0.72 Log_10_ copies/μL, with a positive rate of 33.33% in the vaccine group and 5.90 ± 0.86 Log_10_ copies/μL in the control group, with a positive rate of 100%. At 3 WPC, the average viral load was 0.79 ± 0.63 Log_10_ copies/μL, with a positive rate of 66.67% in the vaccine group, and the control group was 4.90 ± 1.22 Log_10_ copies/μL, with a positive rate of 100%. At 4 WPC, the average viral load was 2.92 ± 0.61 Log_10_ copies/μL, with a positive rate of 100%, and 5.19 ± 0.87 Log_10_ copies/μL in the control group, with a positive rate of 100%. From 1 week after the challenge to the end of the experiment, the serum PCV2 virus dose in the vaccine group was significantly lower than that in the control group (*p* < 0.001) (Figure 4). The area under the PCV2 curve of the vaccine group was also significantly lower than that of the control group (Appendix A).

#### 3.2.6. Pathological Examination

The total scores of gross lesions in the lungs, hilar lymph nodes, inguinal lymph nodes, mesenteric lymph nodes, and kidneys in the vaccine group were lower than those in the control group. A statistical analysis of the Kruskal-Wallis test showed that there was no significant difference between the two groups (Table 2). The results of histopathological lesions were shown in Table 3, and the total scores of histopathological lesions in the lungs, hilar lymph nodes, groin lymph nodes, mesenteric lymph nodes, and kidneys in the vaccine group were also lower than those in the control group.

#### 3.2.7. Immunohistochemical Staining

The results of immunohistochemical staining of the PCV2 antigen are shown in Table 4, and the total scores of hilar lymph nodes, groin lymph nodes, and mesenteric lymph nodes in the vaccine group were also lower than those in the control group.

## 4. Discussion

Porcilis PCV ID is a commercial PCV2 vaccine developed for intradermal use through a needle-free injector (IDAL). The method of administration has no risk of needle breakage and reduces the risk of iatrogenic transmission of disease [19,20]. In our study, we evaluated the safety and efficacy profile of this vaccine with a local PCV2 strain challenge in Taiwan.

We found that this vaccine is safe for use in 3-week-old piglets. Our findings show that even at 5× the normal dose, no adverse reactions were observed. Intradermal vaccination allows for a small dose of 0.2 mL, which is much reduced compared to intramuscular vaccines, and hence improves vaccine safety. Efficacy is preserved with the smaller dose as vaccination into the dermal layer of the skin introduces antigens into the proximity of dendritic cells and dermal lymph nodes. Intradermal swine vaccination has demonstrated similar [34] or improved efficacy [35] compared with the intramuscular route.

In our efficacy study, after the challenge with a local strain of PCV2, all major clinical parameters such as viremia, shedding, mortality, ADG, microscopic and histopathological lesions were statistically significantly different compared to the non-vaccinated control group. Of note, our study challenged pigs at a relatively young age of 7 weeks of age and demonstrated reduction in the clinical impact of PCV2. The prevalence of PCV2 is highest in the grow−finish stage [32,36], and correspondingly the economic impact is the highest as this is the time when pigs are growing the fastest and have the highest potential to lose growth rates [3]. In the field, we expect the results to be even greater if a late finisher stage PCV2 infection occurs.

From a practical perspective, the most important outcome for producers is an improvement in economic parameters such as ADG and FCR [37]. This measurement is not always practical on farms as producers have to individually weigh large numbers of market-age swine to obtain the data for comparison. We propose that other criteria for evaluating the efficacy of PCV2 vaccines exist such as a reduction in the level of viremia, reduction in shedding, and reduction in PCV2-related changes. Additionally, virus levels in tissues [38] can be used. When considering viremia and organ lesions, PCV2 can be distributed throughout the lymphoid tissue, tonsil, spleen, and kidney, contributing to clinical symptoms and tissue damage [9,38]. Therefore, a reduction in viremia is highly indicative of the reduction in damage caused by the PCV2 infection. Increases in viremia levels have also been correlated with decreases in zootechnical parameters [39,40], making viremia an important metric to assess PCV2 vaccine efficacy. Engle et al. [39] found that PCV2 viraemia was an important driver of ADG decline following infection; a moderate negative correlation was observed between the viral load and overall ADG. Our challenge studies indicate that compared to a non-vaccinated challenged control group, the vaccinated pigs displayed statistically significantly lower viremia virus in different tissues and loss of daily weight gain. Interestingly, our results also support the prior work of Engle et al. [39], suggesting that viremia can be used as a proxy for ADG measurement if the farm is not able to measure ADG.

The work of Patterson et al. [21] found that shedding levels of PCV2 after infection were similar across different sample types such as faecal or nasal. Our results were similar. We found that both faecal and nasal viral shedding were reduced in vaccinated pigs, with important implications for biosecurity and control in the farm. In other diseases, such as African Swine Fever [41] and Foot-and-mouth disease [42], shedding through body excretions is an important route for maintaining infection in a susceptible population. The reduction in shedding through vaccination is hence an important tool for long-term control of PCV2. Other authors [43] have also demonstrated the possibility of reducing PCV2 in a farm to undetectable levels by long-term vaccination and reduced shedding.

Our study has at least three limitations: (i) we were not able to grow pigs out to slaughter at 23 weeks of age to ascertain if the immunity granted by the vaccine would last until slaughter; (ii) we also note that there were a small number of pigs in this study, and that even though the vaccine was demonstrated to be safe, further evaluation in field studies is warranted; (iii) we also were not able to test the pigs in a situation with co-infection, especially with the PRRS virus. Other authors have found that lesions of PCV2 are worsened with co-infection, especially with *Mycoplasma hyopneumoniae* and PRRS virus [44]. Future studies will evaluate the field efficacy of this product in a situation with co-infection. In our study, differences between gross lesions and microscopic lesions were observed. For example, microscopic lesions were observed in the kidney of pigs #607, #608, #609, #603, and #619, but gross lesions were not observed. This highlights the importance of microscopic examination of tissues to confirm PCV2-associated disease as gross lesions may be mild or invisible. This also highlights the need to confirm findings by immunohistochemistry or in situ hybridization if doubt exists [45].

## 5. Conclusions

The study results support that a one-dose PCV2 vaccine administered intradermally with a needle-free injector is safe and provides protection when challenged with a field PCV2 strain.

## Figures and Tables

**Figure 1 vaccines-13-00343-f001:**
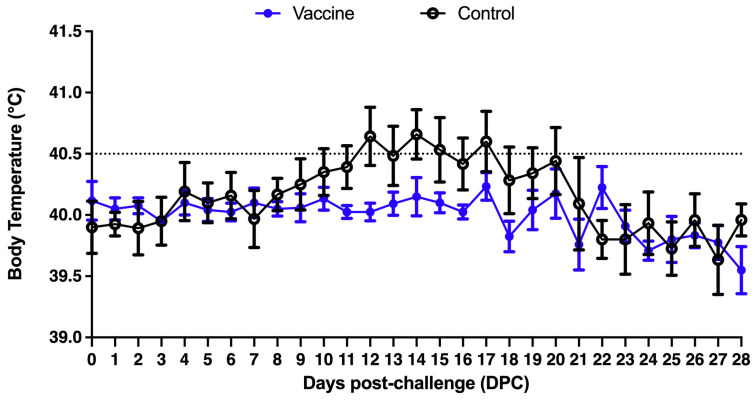
Daily average body temperature after the challenge. The dotted line represents the threshold for a clinical definition of fever (40.5 °C).

**Figure 2 vaccines-13-00343-f002:**
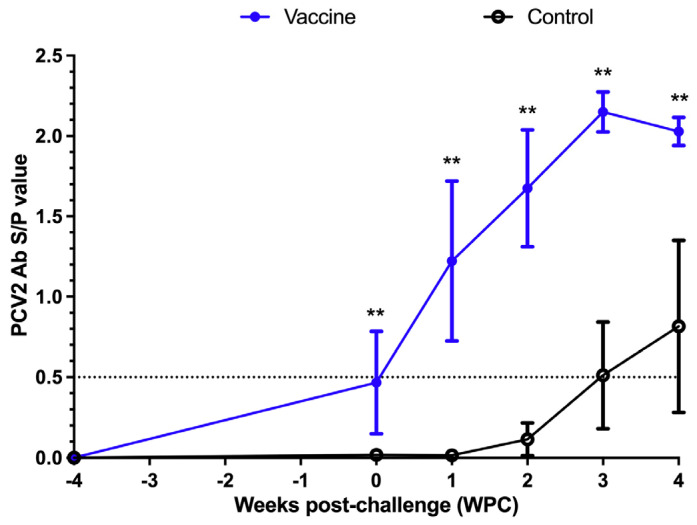
Serodynamic profile of PCV2-specific antibodies measured by BioCheck ELISA. The dotted line is the PCV2 ELISA interpretation threshold (0.5). ** indicates significant differences between different groups at a given date (*p* < 0.01).

**Figure 3 vaccines-13-00343-f003:**
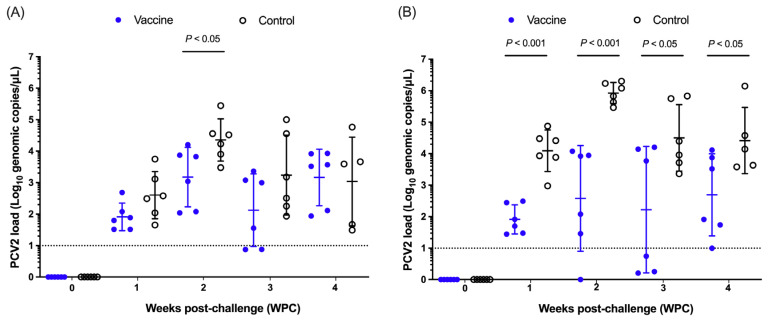
Analysis of PCV2 load in nasal swabs (**A**) and anal swabs (**B**) after the challenge. Blue (closed circle) is the vaccine group, black (open circle) is the control group, the dotted line is the PCV2 qPCR detection limit, and the dotted line below is judged to be negative. *p* < 0.05 indicated a statistically significant difference between the control and vaccine groups.

**Figure 4 vaccines-13-00343-f004:**
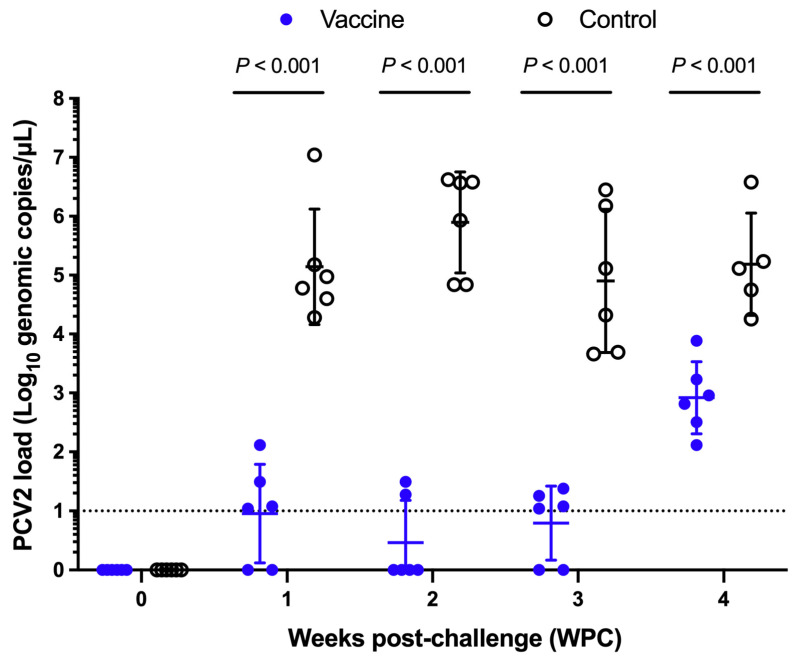
Comparison of the amount of PCV2 in the serum of pigs after the challenge. Blue is the vaccine group, white is the control group, the dotted line is the PCV2 qPCR detection limit, and the dotted line below is judged to be negative. *p* < 0.001 indicated a highly statistical significant difference between the control and vaccine groups.

**Table 1 vaccines-13-00343-t001:** Comparison of average daily gain, mortality, and incidence of stunted pigs during the test period.

Parameter	Weeks of Age	Vaccinated Group	Control Group	*p* Value
Body weight (kg)	3	4.04 ± 0.21	3.93 ± 0.26	0.416
	7	10.91 ± 2.19	11.82 ± 1.53	0.424
	11	21.46 ± 2.80	18.87 ± 4.75	0.276
Average daily weight gain (kg/day)	3–7	0.245	0.282	0.3939
	7–11	0.377	0.250	0.0043
	3–11	0.311	0.267	0.6991
Mortality rate (%)	3–11	0	16.7	
Incidence of poor growth pigs (%)	3–11	0	16.7	

**Table 2 vaccines-13-00343-t002:** Scores of gross pathological lesions of pigs in the vaccine and control groups.

Groups	Ear Tags	Lung	Lymph Node			Kidney
			Hilar	Inguinal	Mesenteric	
Control	605	0.5	1.5	1.5	2	1
	607	1.5	0.5	0	1.5	0
	608	1	1	0.5	1.5	0
	609	2.5	0.5	3	1	0
	610	2.5	0.5	0	0.5	0.5
	611	2	2	0.5	2.5	0
	Total	7.5	6	5.5	9	1.5
Vaccine	612	0	0	1	0.5	0
	613	1	1.5	0	1	0
	615	1	0.5	1	0.5	0
	618	0.5	0.5	0.5	2	0
	619	1	1.5	0.5	2	0
	620	1.5	1	0	1.5	0
	Total	5	5	3	7.5	0
*p* value ^a^		0.092	0.688	0.748	0.575	0.336

^a^ *p* < 0.05 indicated a statistically significant difference between the control and vaccine groups.

**Table 3 vaccines-13-00343-t003:** Scores of microscopic histopathological lesions of pigs in the vaccine and control groups.

Groups	Ear Tags	Lung	Lymph Node			Kidney
			Hilar	Inguinal	Mesenteric	
Control	605	1	1.5	1	3.5	4
	607	1	1	2	2	0.5
	608	1.5	1.5	2.5	1.5	0.5
	609	3	1.5	2	2	3.5
	610	0	1	1	1.5	0
	611	1	1	1.5	1	1
	Total	7.5	7.5	10	11.5	9.5
Vaccine	612	0.5	1	0.5	1	0
	613	0	0.5	0	2	1
	615	0	0	1.5	1.5	0
	618	0	0	0	2	0
	619	0.5	0	0.5	2	1
	620	0	0	0.5	0	0
	Total	1	1.5	3	8.5	2
*p* value ^a^		0.024	0.008	0.013	0.575	0.149

^a^ *p* < 0.05 indicated a statistically significant difference between the control and vaccine groups.

**Table 4 vaccines-13-00343-t004:** Scores of PCV2 antigen immunohistochemical staining in the vaccine and control groups.

Groups	Ear Tags	Lymph Node		
		Hilar	Inguinal	Mesenteric
Control	605	1.5	1.5	2
	607	0	0	1
	608	0	2	0.5
	609	3.5	2	3.5
	610	2.5	2	2.5
	611	0	1	0.5
	Total	7.5	8.5	10
Vaccine	612	0	0.5	0.5
	613	0	0	0
	615	0	0	0
	618	0	0	0
	619	0	0	0
	620	0	1	0
	Total	0	1.5	0.5
*p* value ^a^		0.149	0.030	0.006

^a^ *p* < 0.05 indicated a statistically significant difference between the control and vaccine groups.

## Data Availability

The data presented in this study are available on request from the corresponding author.

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
