# Peer review of "The Evaluation of a Porcine Circovirus Type 2 (PCV2) Intradermal Vaccine Against a PCV2 Field Strain"

_vaccines, 2025, doi:10.3390/vaccines13040343_

Round 1
Reviewer 1 Report
Comments and Suggestions for Authors
The authors evaluated a PCV2 vaccine administered by intradermal route with at needle-free IDAL device. The results demonstrated that the vaccine was safe in a overdose study and that the vaccine was effective and able to reduce PCV2 viral shedding in nasal and anal swabs as well as blood qPCR.
The paper is well written and the results are scientifically sound.
Minor comments.
Table 1 Vaccinated group remove the "d" also "Control"
Line 286 significant
Discussion line 321 It should be noted that there were a small number of animals in this study and that even though the vaccine was demonstrated to be safe further evaluation in field studies should be evaluated.
It would be interesting to know the difference in the weight of the pigs between the vaccinated and control pigs at 11 weeks of age and if this is significant? This could be added to Table 1 and discussed/
Author Response
Reviewer # 1
The authors evaluated a PCV2 vaccine administered by intradermal route with at needle-free IDAL device. The results demonstrated that the vaccine was safe in a overdose study and that the vaccine was effective and able to reduce PCV2 viral shedding in nasal and anal swabs as well as blood qPCR.
The paper is well written and the results are scientifically sound.
Minor comments.
- Table 1 Vaccinated group remove the "d" also "Control"
Our response: We thank the Reviewer # 1 for pointing out this mistake. (Table 1)
- Line 286 significant
Our response: Thank you for your query. We have modified this as suggested. (Line 292)
- Discussion line 321 It should be noted that there were a small number of animals in this study and that even though the vaccine was demonstrated to be safe further evaluation in field studies should be evaluated.
Our response: Thanks for the reviewer # 1’s precise concerns. We have added this limitation in Discussion. (Line 376-378)
- It would be interesting to know the difference in the weight of the pigs between the vaccinated and control pigs at 11 weeks of age and if this is significant? This could be added to Table 1 and discussed/
Our response: We thank the reviewer # 1 for the insightful point. We have updated table 1, there is not significant between the vaccinated and control pigs at 11 weeks of age. (Table 1)

Reviewer 2 Report
Comments and Suggestions for Authors
The manuscript (vaccines-349084) by Cheng-Kai Hsieh et al. entitled “The evaluation of a PCV2 intradermal vaccine against a PCV2 field strain” describes the safety and efficacy of commercial porcine circovirus type 2 vaccine (Porcilis PCV ID, MSD Animal Health). The findings are not scientifically new but provide valuable evidence that the Porcilis PCV ID is effective for the local PCV2 (CYC08 strain) challenge.
This reviewer thinks the findings are worth noting in the journal, Vaccines. However, there are “Major” and “Minor” issues to be clarified before accepting for publication.
Major issues;
- Figures 3, 4, and 5 do not show the results correctly and mislead the understanding. The vertical Y-axis is shown by logarithm. However, a “+/- SD” range does not correspond to the Y-axis scale. In addition, a range of “+/- SD” for vaccinated groups at 2, 3, and 4 WPC of Figure 4 is too narrow if considering the fluctuation of individual pig values. A “+/- SD” range for vaccinated groups at 2 WPC of Figure 5 seems strange, considering that only 33.33% (2/6) is a PCV2 DNA positive (line 273). Please calculate the deviation values again. Moreover, please only show the significant P values but delete the non-significant ones. This reviewer would like to suggest making one composite figure 3 consisting of 3A (nasal secretion) and 3B (anal secretion). Individual PCV2 DNA values in the serum had better be shown as in Figures 3 and 4 to unify the appearance.
- One of six pigs in the control group died during the observation period (probably between 3 and 4 WPC). However, no precise results or explanations are shown in either Tables (except Table 1) or Figures. It needs to be indicated clearly.
- Safety-related issues are described in the Abstract (lines 21-25, 31 and 34-35), Material and Methods (lines 101-107), and Discussion (lines 320-323) sections, but only one sentence is shown in the Results section. Considering the importance of safety issues, results such as the local reaction around the vaccination site, body weight gain, body temperature, and so on should be shown as the obtained results.
Minor issues;
- Lines 2-3, Title: Abbreviated or short forms are not allowed in the title except gene or protein names (See, Instruction for Authors). PCV2 should be spelled like “Porcine Circovirus Type 2 (PCV2)”.
- Lines 40-41: The first sentence had better be described as “Porcine circovirus type 2 (PCV2) is a major etiological agent of the Porcine Respiratory Disease Complex (PRDC) [1].”.
- Lines 41-43: The genus names, Circovirus and Cyclovirus, should be italicized.
- Lines 40-46: Genotype of PCV2 had better be explained in this Introduction section. PCV2 is currently classified into eight different genotypes, designated PCV2a to 2h. After the PCV2a vaccine was introduced in 2007, PCV2b had overtaken PCV2a as the predominant one. PCV2b and PCV2d are globally distributed, while others are detected in limited areas. It may be worth describing that PCV2d has become more prevalent than PCV2b worldwide, but that PCV2d does not escape the immunity induced by PCV2a-based vaccines with the proper references.
- Line 48: Please delete hyphen from “multisys-temic”.
- Lines 50 and 55: The words “industrial pork production” and “producers” had better be described as “pig-producing industry”.
- Line 61: The word “farm” had better be described as “the pig farm”.
- Line 85: The term “adaptive and humoral responses” had better be described as “adaptive cellular and humoral immune responses” or “cellular and humoral immune responses”.
- Lines 82: The meaning of “Asian context” is not clear. Does it mean the local PCV2 strains are prevalent in Asia? Please add evidence if there is a unique feature of PVC2 in Asia to clarify the meaning.
- Lins101-107: Please indicate the weeks of age of four piglets used for the safety study and the supplier from which authors obtained the SPF piglets. Did the authors prepare the quarantine term before using piglets?
- Lines 105-106: Please clarify what criteria the authors set for monitoring the adverse events.
- Line 112: The words “sham vaccinated with 0.2 ml of normal saline” had better be described as “sham vaccinated with 0.2 ml of saline as a control group”.
- Line 113: Please indicate the origin and the genotype of the PCV2 CYC08 strain.
- Line 120: The term “the antibody price difference” had better be described as “the antibody S/P value” according to the Y-axis of Figure 2.
- Line 131: Please indicate weeks or days when the nasal and anal samples were taken. In addition, please indicate the method by which PCV2 DNA was extracted from the sample fluids.
- Line 139: The first sentence had better be modified to show that blood samples were collected from pigs. In addition, please indicate the method by which PCV2 DNA was extracted from the serum.
- Line 161: The meaning of detoxification is not clear. Does it mean the shedding? Please clarify the meaning.
- Lines 169-175: The meaning of a long sentence is unclear. Please modify the sentence to clarify the meaning.
- Line 189: Please insert a period between “challenge” and “the dotted line”.
- Lines 199-203: It is described that “At 2 WPC”, “the pigs in the control group began to turn positive”. However, the description is inconsistent with the obtained data because all control sera at 2 WPC are below the threshold line, indicating seronegative to PCV2 Ab. (Figure 2).
- Lines 210-213: The meaning of a long sentence is unclear. Please modify the sentence to clarify the meaning.
- Lines 218-219: A sentence “During the trial, the mortality rate of the control group was 16.67%, and the mortality rate of the vaccine group was 0.” had better be modified according to the description of Table 1, such as “During the trial, the mortality rate of the control group was 16. 7% (1/6), and that of the vaccine group was 0% (0/6).”.
- Line 225: Please check whether “PCV2 antigen” is correct or not. “PCV2 genome” or “PCV2 DNA”?
- Lines 240-241, Figure 3 caption: “Blue is the vaccine group, white is the control group” had better be indicated as “Blue (closed circle) is the vaccine group, black (open circle) is the control group”.
- Lines 288-290, Table 2: Tables 2 seem unnecessary in the main text. Please consider showing it as the supplemental results.
- Line 299: Please modify the Table title. “Scores of gross pathological lesions” is much better to contrast with Table 4.
- Lines 299-305, Tables 3 and 4: Please explain the meaning of the underline and the bold character for pig number #605.
- Lines 299-305, Table 3 and 4: For example, gross lesions (scores 2.5, 1, and 0.5) are found in the lungs of #610, #615, and #618 pigs, but none of the microscopic lesions are found in the same lungs. Vise versa, microscopic lesions (scores 0.5, 0.5, 3.5, 1, and 1) are found in the kidney of #607, #608, #609, #603, and #619 pigs, but none of the gross lesions are found in the same kidney. Please explain the discrepancy between gross lesions and macroscopic lesions.
- Line 302: Please modify the Table title. “Scores of microscopic histopathological lesions” is much better to contrast with Table 3.
- Line 339: The term “feed conversion ratio” should be “FCR” because it has already been defined in line 46.
Comments on the Quality of English Language
Some comments related to the quality of English are included in the Minor issues listed in the separate repot for the authors.
Author Response
Reviewer # 2
The manuscript (vaccines-349084) by Cheng-Kai Hsieh et al. entitled “The evaluation of a PCV2 intradermal vaccine against a PCV2 field strain” describes the safety and efficacy of commercial porcine circovirus type 2 vaccine (Porcilis PCV ID, MSD Animal Health). The findings are not scientifically new but provide valuable evidence that the Porcilis PCV ID is effective for the local PCV2 (CYC08 strain) challenge.
This reviewer thinks the findings are worth noting in the journal, Vaccines. However, there are “Major” and “Minor” issues to be clarified before accepting for publication.
Major issues;
- Figures 3, 4, and 5 do not show the results correctly and mislead the understanding. The vertical Y-axis is shown by logarithm. However, a “+/- SD” range does not correspond to the Y-axis scale. In addition, a range of “+/- SD” for vaccinated groups at 2, 3, and 4 WPC of Figure 4 is too narrow if considering the fluctuation of individual pig values. A “+/- SD” range for vaccinated groups at 2 WPC of Figure 5 seems strange, considering that only 33.33% (2/6) is a PCV2 DNA positive (line 273). Please calculate the deviation values again. Moreover, please only show the significant P values but delete the non-significant ones. This reviewer would like to suggest making one composite figure 3 consisting of 3A (nasal secretion) and 3B (anal secretion). Individual PCV2 DNA values in the serum had better be shown as in Figures 3 and 4 to unify the appearance.
Our response: Thank you for your suggestion. We have recalculated the Standard Deviation values again and confirm they are correct. We have deleted the non-significant P values and only display now the significant P values. We have made one composite Figure 3 combining 3A and 3B. We hope that we have the changes as you have requested but please let us know if we have misunderstood anything. (Line 275-280)
- One of six pigs in the control group died during the observation period (probably between 3 and 4 WPC). However, no precise results or explanations are shown in either Tables (except Table 1) or Figures. It needs to be indicated clearly.
Our response: Thank you for your suggestion. We have added this into Section 3.2.3 (Line 225-230)
- Safety-related issues are described in the Abstract (lines 21-25, 31 and 34-35), Material and Methods (lines 101-107), and Discussion (lines 320-323) sections, but only one sentence is shown in the Results section. Considering the importance of safety issues, results such as the local reaction around the vaccination site, body weight gain, body temperature, and so on should be shown as the obtained results.
Our response: Thank you for your suggestion. We have added this into Section 3.1 (Line 190-193)
Minor issues;
- Lines 2-3, Title: Abbreviated or short forms are not allowed in the title except gene or protein names (See, Instruction for Authors). PCV2 should be spelled like “Porcine Circovirus Type 2 (PCV2)”.
Our response: Thank you for your query. We have modified this as suggested. (Line 2-3)
- Lines 40-41: The first sentence had better be described as “Porcine circovirus type 2 (PCV2) is a major etiological agent of the Porcine Respiratory Disease Complex (PRDC) [1].”.
Our response: We have modified this as suggested. (Line 42-43)
- Lines 41-43: The genus names, Circovirus and Cyclovirus, should be italicized.
Our response: We have modified this as suggested. (Line 44-45)
- Lines 40-46: Genotype of PCV2 had better be explained in this Introduction section. PCV2 is currently classified into eight different genotypes, designated PCV2a to 2h. After the PCV2a vaccine was introduced in 2007, PCV2b had overtaken PCV2a as the predominant one. PCV2b and PCV2d are globally distributed, while others are detected in limited areas. It may be worth describing that PCV2d has become more prevalent than PCV2b worldwide, but that PCV2d does not escape the immunity induced by PCV2a-based vaccines with the proper references.
Our response: Thank you for this suggestion. We have modified our text accordingly. (Line 57-60)
- Line 48: Please delete hyphen from “multisys-temic”.
Our response: We have modified this as suggested. (Line 49)
- Lines 50 and 55: The words “industrial pork production” and “producers” had better be described as “pig-producing industry”.
Our response: We have modified this as suggested. (Line 51-52, 54)
- Line 61: The word “farm” had better be described as “the pig farm”.
Our response: We have modified this as suggested. (Line 66)
- Line 85: The term “adaptive and humoral responses” had better be described as “adaptive cellular and humoral immune responses” or “cellular and humoral immune responses”.
Our response: We have modified this as suggested. (Line 90-91)
- Lines 82: The meaning of “Asian context” is not clear. Does it mean the local PCV2 strains are prevalent in Asia? Please add evidence if there is a unique feature of PVC2 in Asia to clarify the meaning.
Our response: Thank you for your suggestion. We have added additional lines to clarify our meaning (Line 98)
- Lins101-107: Please indicate the weeks of age of four piglets used for the safety study and the supplier from which authors obtained the SPF piglets. Did the authors prepare the quarantine term before using piglets?
Our response: We have modified this as suggested. (Line 110)
- Lines 105-106: Please clarify what criteria the authors set for monitoring the adverse events.
Our response: We have modified this as suggested. (Line 114-116)
- Line 112: The words “sham vaccinated with 0.2 ml of normal saline” had better be described as “sham vaccinated with 0.2 ml of saline as a control group”.
Our response: We have modified this as suggested. (Line 122-123)
- Line 113: Please indicate the origin and the genotype of the PCV2 CYC08 strain.
Our response: Thank you for your query. We have added the information about PCV2 CYC08. (Line 125)
- Line 120: The term “the antibody price difference” had better be described as “the antibody S/P value” according to the Y-axis of Figure 2.
Our response: We have modified this as suggested. (Line 132)
- Line 131: Please indicate weeks or days when the nasal and anal samples were taken. In addition, please indicate the method by which PCV2 DNA was extracted from the sample fluids.
Our response: Thank you for your query. We have added the information about sampling timing and DNA preparation. (Line 146-149)
- Line 139: The first sentence had better be modified to show that blood samples were collected from pigs. In addition, please indicate the method by which PCV2 DNA was extracted from the serum.
Our response: We have modified this as suggested. (Line 154-155)
- Line 161: The meaning of detoxification is not clear. Does it mean the shedding? Please clarify the meaning.
Our response: Thank you for your query. We have updated the word to “shedding”. (Line 177)
- Lines 169-175: The meaning of a long sentence is unclear. Please modify the sentence to clarify the meaning.
Our response: Thank you for your kind comments and suggestion. We have rewritten the sentence. (Line 185-186)
- Line 189: Please insert a period between “challenge” and “the dotted line”.
Our response: We have modified this as suggested. (Line 203)
- Lines 199-203: It is described that “At 2 WPC”, “the pigs in the control group began to turn positive”. However, the description is inconsistent with the obtained data because all control sera at 2 WPC are below the threshold line, indicating seronegative to PCV2 Ab. (Figure 2).
Our response: We thank the Reviewer # 2 for pointing out this issue. We have rewritten the sentence. (Line 213-215)
- Lines 210-213: The meaning of a long sentence is unclear. Please modify the sentence to clarify the meaning.
Our response: Thank you for your kind comments and suggestion. We have substantially rewritten the sentence.(Line 225-230)
- Lines 218-219: A sentence “During the trial, the mortality rate of the control group was 16.67%, and the mortality rate of the vaccine group was 0.” had better be modified according to the description of Table 1, such as “During the trial, the mortality rate of the control group was 16. 7% (1/6), and that of the vaccine group was 0% (0/6).”.
Our response: We have modified this as suggested. (Line 235-236)
- Line 225: Please check whether “PCV2 antigen” is correct or not. “PCV2 genome” or “PCV2 DNA”?
Our response: We have modified this as suggested. (Line 240)
- Lines 240-241, Figure 3 caption: “Blue is the vaccine group, white is the control group” had better be indicated as “Blue (closed circle) is the vaccine group, black (open circle) is the control group”.
Our response: We have modified this as suggested. (Line 276-277)
- Lines 288-290, Table 2: Tables 2 seem unnecessary in the main text. Please consider showing it as the supplemental results.
Our response: We have modified this as suggested. Table 2 moves to supplemental table 1. (Line 294-295)
- Line 299: Please modify the Table title. “Scores of gross pathological lesions” is much better to contrast with Table 4.
Our response: We have modified this as suggested. (Line 309)
- Lines 299-305, Tables 3 and 4: Please explain the meaning of the underline and the bold character for pig number #605.
Our response: This is our mistake and we have removed it accordingly. There is no meaning for the underline or bold character in pig #605. (Line 312-313)
- Lines 299-305, Table 3 and 4: For example, gross lesions (scores 2.5, 1, and 0.5) are found in the lungs of #610, #615, and #618 pigs, but none of the microscopic lesions are found in the same lungs. Vise versa, microscopic lesions (scores 0.5, 0.5, 3.5, 1, and 1) are found in the kidney of #607, #608, #609, #603, and #619 pigs, but none of the gross lesions are found in the same kidney. Please explain the discrepancy between gross lesions and macroscopic lesions.
Our response: We have modified our manuscript accordingly to explain this in the Discussion section. (Line 381-391)
- Line 302: Please modify the Table title. “Scores of microscopic histopathological lesions” is much better to contrast with Table 3.
Our response: We have modified this as suggested. (Line 311)
- Line 339: The term “feed conversion ratio” should be “FCR” because it has already been defined in line 46.
Our response: We have modified this as suggested. (Line 346)

Round 2
Reviewer 2 Report
Comments and Suggestions for Authors
The manuscript (vaccines-349084) by Cheng-Kai Hsieh et al. entitled “The evaluation of a Porcine Circovirus Type 2 (PCV2) intradermal vaccine against a PCV2 field strain” was revised according to the suggestions and comments by the reviewers. The authors succeeded in improving the manuscript for publication in Vaccines.
However, “Minor” issues still remain to be clarified before accepting for publication.
Minor issues;
- Lines 21, 36, 97, and 382: Authors claim in the manuscript that the PCV2a CYC08 strain is a contemporary Asian strain. This reviewer understands that the PCV2a CYC08 strain was initially isolated in Taiwan and was used as the challenge virus, but suggests showing in the manuscript the apparent results that support CYC08-like strains are prevalent in Asian regions, to say a contemporary Asian strain.
- Lines 205-207: It is described in the text that “Before the challenge at 7 weeks of age (4 weeks after immunization or 0 week post-challenge (WPC)), the PCV2-specific antibodies in the immunization group were shown positive”. However, the mean PCV2 Ab S/P value at 0 WPC in Figure 2 is below the S/P threshold line, indicating that most pig anti-PCV2 sera were still negative. Please describe the results more appropriately.
- Lines 275-283: The mean viral loads of the vaccinated group are shown as 1.39+/-0.15 (2 WPC), 1.19+/-0.16 (3 WPC), 2.92+/-0.61 Log10 copies/µL (4 WPC). However, the viral load graph (Figure 4) does not show the results correctly and mislead the understanding. Both the copy number and the “+/- SD” range shown in the figure do not correspond to the description in the manuscipt.
- Line 291; “P < 0.01” had better be shown as “P< 0.001”.Lines 416, 521, and 526: The style of references does not meet the journal style. Please do not omit page range.
- Lines 460, 475, 478, 481, 485, 489, 502, 510, and 520: The style of references does not meet the journal style. All authors should be listed. Please do not use “et al.”.
- Lines 460, 475, 478, 481, 485, 489, 502, 510, and 520: The style of references does not meet the journal style. All authors should be listed. Please do not use “et al.”.
Several sentences begin with Arabic numerals, so they should be spelled out. Otherwise, the sentences need to be modified so that the numbers are not at the beginning of the sentence.
Author Response
Minor issues;
1. Lines 21, 36, 97, and 382: Authors claim in the manuscript that the PCV2a CYC08 strain is a contemporary Asian strain. This reviewer understands that the PCV2a CYC08 strain was initially isolated in Taiwan and was used as the challenge virus, but suggests showing in the manuscript the apparent results that support CYC08-like strains are prevalent in Asian regions, to say a contemporary Asian strain.
Our response: Thank for the Reviewer # 2’s precise concerns. We have deleted “contemporary Asian strain” in text.(Line 21, 37, 97, 382)
2. Lines 205-207: It is described in the text that “Before the challenge at 7 weeks of age (4 weeks after immunization or 0 week post-challenge (WPC)), the PCV2-specific antibodies in the immunization group were shown positive”. However, the mean PCV2 Ab S/P value at 0 WPC in Figure 2 is below the S/P threshold line, indicating that most pig anti-PCV2 sera were still negative. Please describe the results more appropriately.
Our response: We have modified this as suggested. (Line 207)
3. Lines 275-283: The mean viral loads of the vaccinated group are shown as 1.39+/-0.15 (2 WPC), 1.19+/-0.16 (3 WPC), 2.92+/-0.61 Log10 copies/µL (4 WPC). However, the viral load graph (Figure 4) does not show the results correctly and mislead the understanding. Both the copy number and the “+/- SD” range shown in the figure do not correspond to the description in the manuscipt.
Our response: We thank the Reviewer # 2 for pointing out this issue. We have recheck and updated the data. (Line 276, 278, 280)
4. Line 291; “P < 0.01” had better be shown as “P< 0.001”.
Our response: We have modified this as suggested. (Line 291)
5. Lines 416, 521, and 526: The style of references does not meet the journal style. Please do not omit page range.
Our response: We thank the Reviewer # 2 for pointing out this issue. We have updated the style of reference.
6. Lines 460, 475, 478, 481, 485, 489, 502, 510, and 520: The style of references does not meet the journal style. All authors should be listed. Please do not use “et al.”.
Our response: We thank the Reviewer # 2 for pointing out this issue. We have updated the style of reference.